# Association of Visual-Based Signals with Electroencephalography Patterns in Enhancing the Drowsiness Detection in Drivers with Obstructive Sleep Apnea

**DOI:** 10.3390/s24082625

**Published:** 2024-04-19

**Authors:** Riaz Minhas, Nur Yasin Peker, Mustafa Abdullah Hakkoz, Semih Arbatli, Yeliz Celik, Cigdem Eroglu Erdem, Beren Semiz, Yuksel Peker

**Affiliations:** 1College of Engineering, Koc University, Istanbul 34450, Turkey; rminhas22@ku.edu.tr (R.M.); besemiz@ku.edu.tr (B.S.); 2Department of Mechatronics Engineering, Sakarya University of Applied Sciences, Sakarya 54050, Turkey; yasinpeker@subu.edu.tr; 3Graduate School of Computer Engineering, Istanbul Technical University, Istanbul 34469, Turkey; mustafa.hakkoz@gmail.com; 4Graduate School of Health Sciences, Koc University, Istanbul 34010, Turkey; sarbatli20@ku.edu.tr; 5Research Center for Translational Medicine (KUTTAM), Koc University, Istanbul 34010, Turkey; yecelik@ku.edu.tr; 6Department of Electrical and Electronics Engineering, Ozyegin University, Istanbul 34794, Turkey; cigdem.erogluerdem@gmail.com; 7Department of Pulmonary Medicine, School of Medicine, Koc University, Istanbul 34010, Turkey; 8Sahlgrenska Academy, University of Gothenburg, 40530 Gothenburg, Sweden; 9School of Medicine, Lund University, 22185 Lund, Sweden; 10School of Medicine, University of Pittsburgh, Pittsburgh, PA 15213, USA

**Keywords:** CLOSDUR, discrete wavelet transform, driving simulator, drowsiness, electroencephalography, image processing, obstructive sleep apnea, PERCLOS

## Abstract

Individuals with obstructive sleep apnea (OSA) face increased accident risks due to excessive daytime sleepiness. PERCLOS, a recognized drowsiness detection method, encounters challenges from image quality, eyewear interference, and lighting variations, impacting its performance, and requiring validation through physiological signals. We propose visual-based scoring using adaptive thresholding for eye aspect ratio with OpenCV for face detection and Dlib for eye detection from video recordings. This technique identified 453 drowsiness (PERCLOS ≥ 0.3 || CLOSDUR ≥ 2 s) and 474 wakefulness episodes (PERCLOS < 0.3 and CLOSDUR < 2 s) among fifty OSA drivers in a 50 min driving simulation while wearing six-channel EEG electrodes. Applying discrete wavelet transform, we derived ten EEG features, correlated them with visual-based episodes using various criteria, and assessed the sensitivity of brain regions and individual EEG channels. Among these features, theta–alpha-ratio exhibited robust mapping (94.7%) with visual-based scoring, followed by delta–alpha-ratio (87.2%) and delta–theta-ratio (86.7%). Frontal area (86.4%) and channel F4 (75.4%) aligned most episodes with theta–alpha-ratio, while frontal, and occipital regions, particularly channels F4 and O2, displayed superior alignment across multiple features. Adding frontal or occipital channels could correlate all episodes with EEG patterns, reducing hardware needs. Our work could potentially enhance real-time drowsiness detection reliability and assess fitness to drive in OSA drivers.

## 1. Introduction

Obstructive sleep apnea (OSA) is a common condition presenting with snoring, recurrent breathing pauses during sleep, disturbances in oxygenation, and frequent arousals during sleep [1]. These disruptions result in symptoms like fatigue and excessive daytime sleepiness (EDS), significantly impacting attention and cognitive functions, especially during driving tasks [2,3]. OSA affects around 2% to 4% of women and 4% to 11% of men in the middle-aged population [4]. OSA patients face a notably higher risk of motor vehicle accidents, with rates two to seven times higher compared to the general population [5,6]. A specific study highlighted the increased accident risk in individuals with OSA compared to those without OSA (crash relative risk = 2.43, 95% CI: 1.21–4.89, *p* = 0.013) [7]. The US National Highway Traffic Safety Administration (NHTSA) estimates that over 100,000 road accidents occur annually due to drowsiness, resulting in 800 fatalities and 50,000 injuries [8]. Given these statistics, detecting drowsiness in OSA patients is critical for road safety and assessing their driving fitness.

While the Multiple Sleep Latency Test (MSLT) and Maintenance of Wakefulness Test (MWT) are established in sleep medicine [9,10] for measuring daytime drowsiness, their limitations arise from operating in non-interactive, sleep-inducing settings. These restrictions limit their ability to mimic real-world driving conditions, hindering the accurate evaluation of a driver’s readiness. The multifaceted nature of driving involves perceptual, motor, and cognitive abilities [11], making these tests inadequate for assessing drowsiness effectively.

Simulated driving environments are also an established method for detecting drowsiness, proving highly effective in replicating real-world scenarios [12,13,14]. These environments encompass three primary data classifications: (1) physiological-based signals [15,16,17], comprising EEG, EOG, EMG, and ECG; (2) behavior-based signals such as eye movement [18,19], yawning [20], and head-nodding [21]; and (3) vehicle-based signals, which include lane deviation, steering entropy, and out-of-road events [22,23]. Research highlights EEG measurements as highly effective in promptly identifying drowsiness onset, surpassing both behavior-based and vehicle-based systems [24,25,26]. Although, behavior-based system lags subtly behind EEG measures in identifying drowsiness onset, detecting early signs such as eye-blinking linked to drowsiness before any lateral vehicle displacement occurs [27]. Since vehicle-based systems issue alerts later in the initial drowsiness phase, potentially limiting accident prevention opportunities, relying solely on this technique is not advisable. Instead, combining it with other methods to detect a driver’s drowsiness proves to be more effective [24].

PERCLOS, a behavior-based signal approved by NHTSA for drowsiness detection independently, measures the duration of eyes at least 80 percent closed within a minute [28]. It can be calculated using built-in algorithms in eye-tracking systems (ETSs) like SmartEye [27] or via image processing from recorded facial videos [18,19]. However, limitations in image processing include challenges with video/image quality, eyewear interference, varying lighting, and head movement, impacting performance [24,26,27]. Addressing these constraints is crucial, as relying solely on this technique may lack reliability. Hence, validating it against established physiological signals and mitigating the limitations tied to it are essential for enhanced dependability in real-world scenarios.

### 1.1. Related Works

Previous research identified drowsiness via PERCLOS, albeit with limitations. One study [18] achieved 88.9% precision by employing skin color identification, Sohel edge operator for eye localization, and dynamic templates for eye tracking. Additionally, another study [19] developed a real-time application utilizing the Viola–Jones detector, achieving 90% accuracy through nearest neighbor IBk and J48 decision tree algorithms. Several studies have integrated PERCLOS with additional behavioral parameters, such as average eye closure speed [29], head movement [30], and yawning episodes [31], enhancing the effectiveness of drowsiness detection. Furthermore, study [27] correlated PERCLOS-based drowsiness with neural patterns, revealing increased theta and delta powers as PERCLOS escalated during driving tests. Study [28] utilized photoplethysmography imaging (PPGI) to derive heart rate variability (HRV) and LF/HF ratio, achieving 92.5% accuracy by correlating these HRV-derived parameters with PERCLOS measurements. Moreover, a couple of studies [32,33] integrated PERCLOS with vehicle-based signals, such as steering wheel movement [32] and lane position [33], while another [34] merged PERCLOS with a galvanic skin response (GSR) sensor using Multi-Task Cascaded Convolutional Neural Networks (MTCNNs), effectively predicting the driver’s transition from an awake to a drowsy state at 91% efficacy.

### 1.2. Limitations in Previous Studies and a Proposed Solution

Study [18] considered a 250-millisecond eyeblink indicative of drowsiness, despite normal blinks lasting between 100–400 milliseconds [35]. Studies [18,19] exclusively relied on PERCLOS for drowsiness detection. Studies [29,30,31] did not validate their drowsiness detection systems, which integrated PERCLOS with other behavioral parameters, against any physiological signals. Study [27] used SmartEye for PERCLOS calculation, noted as reliable but cost ineffective. A study [28] obtained PPGI from facial images, showing a lower accuracy compared to conventional PPG. Studies [32,33] attempted to correlate PERCLOS with vehicle-based signals, but they did not address the potential time lag between the two signals. Limitations in calculating PERCLOS via image processing were unaddressed in all studies except one [27]. Furthermore, none of the studies established a one-to-one correlation between visual-based scoring from PERCLOS and physiological signals; they examined their general association. Lastly, no one identified EEG channel or brain region sensitivity when correlating visual-based scoring with EEG patterns.

In our study, we aim to enhance drowsiness detection reliability using a driving simulator in clinically diagnosed OSA patients. Our approach involves adopting adaptive thresholding for calculating eye aspect ratio (EAR) to minimize limitations related to PERCLOS computation via image processing. Additionally, we seek to validate this method by establishing a direct correlation between episodes of visual-based scoring and EEG patterns, leveraging ten distinct features. Furthermore, we evaluate the sensitivity of individual EEG channels and brain regions in producing this correlation. Through these steps, our approach effectively addresses the limitations encountered in prior studies. Thus, the major contributions of this paper are as follows:
1.Introducing a visual-based scoring method to detect episodes of drowsiness and wakefulness using adaptive thresholding—instead of fixed thresholding—for eye aspect ratio computation. This method leverages OpenCV for face detection and Dlib for eye region extraction (Section 2.4 and Section 3.1).2.Proposing an integrated approach that correlates visual-based scoring with EEG patterns using ten distinct features to enhance the reliability of drowsiness detection (Section 2.5 and Section 3.1).3.Computing the sensitivity of various EEG channels and brain regions to determine the optimal electrode count for this correlation, leading to minimizing hardware requirements, enhancing wearable applications, and prioritizing user comfort. (Section 2.6 and Section 3.2).


## 2. Materials and Methods

### 2.1. Experimental Setup

The experimental setup utilized in this study (Figure 1) was developed in our prior research [36], encompassing the XBUS PRO Driver Training Simulator (DTS) by ANGRUP Co, Istanbul, Turkey, the NOX-A1 EEG system from Nox Medical Inc. in Reykjavik, Iceland, and a 1080p camera as its core components. The DTS offered both manual and automatic transmission options, equipped with sensors monitoring throttle and brake usage, road deviations, steering irregularities, and potential accidents. Housed within a soundproof cabin, it provided a controlled environment with a constant temperature of 22 °C for all participants and a wide 135-degree field of view. Using ANGRUP Software Technologies (version: Professional 5.2.3), the simulator replicated diverse road conditions such as straight stretches, circular tracks, curved paths, and low-traffic highways to simulate various driving scenarios. The EEG device, featuring 6 channels (Frontal: F4 and F3, Central: C4 and C3, Occipital: O1 and O2) and operating at a sampling rate of 200 Hz, captured neural activity based on the standardized 10–20 electrode placement for consistent positioning [37]. Simultaneously, the dome camera recorded the driver’s facial expressions at a rate of 30 frames per second.

### 2.2. Study Population and Subject Demographics

Previous studies have shown a significantly elevated prevalence of OSA in heavy vehicle professional drivers (42.2%) compared to the general population (5%) [38]. Furthermore, OSA prevalence in men (14–50%) is higher than in women (5–23%) [39,40]. In our study, we used a bus driving simulator with professional drivers. It was conducted in Turkey, where bus driving is male dominated, resulting in the exclusive recruitment of male participants. Therefore, we recruited fifty professional male drivers diagnosed with OSA based on their previous night’s polysomnography results (apnea–hypopnea index [AHI] ≥ 5.0 events/h) from the Sleep Laboratory for a simulator-assisted visual-based drowsiness detection [41]. Table 1 presents the demographics of the subjects. The study protocol was approved by the Koç University Committee on Human Research (2020.292.IRB2.083; 19 June 2020), and a written informed consent was obtained from all participants. Participants with no acute illness were included. Additionally, participants were advised to abstain from consuming caffeinated beverages, such as coffee and energy drinks, as well as other stimulants for 24 h preceding the experiment [42]. 

### 2.3. Experimental Design

Participants meeting eligibility criteria underwent a simulated driving session scheduled between 08:00 a.m. and 10:00 a.m., aligning with research indicating increased risks during nighttime or early morning driving hours [43]. Ahead of the experiment, a 10 min training session familiarized drivers with vehicle controls and various driving scenarios, aiming to prepare them for the simulated tasks and improve their performance. Additionally, the interior lights of the cabin were turned off to mimic real-world driving conditions. During the experiment, drivers engaged in a fifty-minute simulated driving session on a two-way highway, maintaining low traffic density, and not exceeding a maximum allowable speed of 62 mph (80 km/h). Figure 2 presents the experimental design used in this study. They wore a 6-channel EEG electrode setup to record their neural activity, and a frontal camera captured their facial expressions, with instructions to abate head movements. 

### 2.4. Data Acquisition

#### 2.4.1. Video-Based Data Acquisition and Visual-Based Scoring

In our study, we employed visual-based scoring—an established technique for detecting drowsiness—by capturing facial video recordings of drivers engaged in simulated driving. Initially, we utilized Python-based OpenCV (version: 2.4.9) library to detect faces in facial video recordings [44]. Detected faces then underwent facial detection procedure using the Dlib library, allowing the estimation of landmark positions on each detected face [45]. The Dlib library’s pre-trained face landmark detector provided coordinates for 68 points, encompassing regions around the eyes, eyebrows, mouth, nose, and chin, as depicted in Figure 3a. Next, the eye aspect ratio (*EAR*) was computed using the Euclidean distance between the identified eye landmarks (see Figure 3b), as outlined in Equation (1).
(1)$$EAR=\frac{\left|P_{2}\;-\;P_{6}\right|+\left|P_{3}\;-\;P_{5}\right|}{2\left|P_{1}-P_{4}\right|}$$

The EAR value typically maintains stability when the eye is open but drops to zero during a blink. Past studies have proposed different EAR thresholds, suggesting that values below 0.28 [46], 0.25 [47], 0.20 [48], 0.18 [49], and 0.16 [50] indicate eye blinking or closure. However, utilizing a fixed threshold value across different individuals, varied lighting conditions, and eyewear presence does not yield precise EAR calculations. Momentary facial expressions like yawning or smiling, as well as head rotations, underscore the necessity for an adaptive threshold value that accommodates diverse individuals and situational factors. The adaptive threshold incorporated a median filter to attenuate abrupt changes in EAR values, effectively reducing noise. Next, employing a moving average filter ensured smoother transitions in EAR values over time, mitigating the impact of environmental variations. Subsequently, the threshold value underwent dynamic readjustment by subtracting a constant value (0.04) following the application of the median filter (of length 17) and moving average filter (of length 5). The filter parameters were determined experimentally. This iterative process continuously refined the threshold based on updated EAR values, significantly improving the precision and adaptability of our technique. Figure 4 illustrates the steps of the adaptive threshold method and compares it with the fixed threshold (EAR = 0.2, suggested by [48]). It highlights the successful detection of eye blinks marked with green ellipses by the adaptive threshold method, contrasting instances missed by the fixed threshold.

After employing adaptive thresholding to detect eye blinks throughout the driving period, we computed two metrics: PERCLOS, indicating the ratio of the number of frames with closed eyes to the total number of frames with both closed and open eyes, and CLOSDUR, measuring the duration of eye closure. Utilizing an established criteria for drowsiness (PERCLOS ≥ 0.3 or CLOSDUR ≥ 2 s) and wakefulness (PERCLOS < 0.3 and CLOSDUR < 2 s) episodes [19], we identified a total of 927 instances encompassing both drowsiness (*n* = 453) and wakefulness (*n* = 474) events throughout the entire driving period. These instances were saved in a CSV file along with their respective timestamps.

#### 2.4.2. Physiological Signal-Based Data Acquisition

EEG signals from all six channels were recorded in the European Data Format (EDF) using the Noxturnal software (version: 6.3.1.34324), a specialized tool designed for recording, analyzing, and processing various physiological data types, including EEG signals [51]. Along with EEG data, temporal information including the start time and end time of the recording was also noted. 

### 2.5. Concurrent Analysis for Validating Visual-Based Scoring with EEG Patterns

Although visual-based scoring is a recognized method for detecting drowsiness [18,19], the inherent limitations in accurately computing PERCLOS emphasize the need to validate this technique using physiology-based signals. To bolster the precision and reliability of visual-based scoring, we integrated data from drowsiness and wakefulness events, obtained through visual-based scoring, with synchronous EEG patterns. This integration aimed to establish a meaningful correlation between these two metrics. To achieve this goal, we developed a customized MATLAB (R2022b) program with specific features tailored to identify associations between visual-based scoring and EEG patterns.

#### 2.5.1. Filtering the Data

We designed finite impulse response (FIR) filters, both high-pass and low-pass, with an order of 25 using the equiripple design method to balance filter complexity and achieve sharpness in frequency response transition regions [36]. This design aimed at optimal attenuation in the stopband while preserving passband characteristics essential for EEG analysis. The high-pass filter, with a cutoff frequency of 1 Hz, effectively attenuates low-frequency artifacts (0.17–0.24 Hz range) commonly caused by eye blinks [52]. Conversely, the low-pass filter, with a cutoff frequency of 30 Hz, was intended to exclude high-frequency noise and potential artifacts originating from electromyogram (EMG) signals [53]. 

#### 2.5.2. Loading and Processing CSV File

The pertinent columns indicating the onset and cessation times for visual-based drowsiness and wakefulness episodes were extracted for analysis. In between two consecutive wakefulness events, there was a drowsiness episode, where the start time of the drowsiness event coincided with the end time of the previous wakefulness episode, and the end time of the drowsiness event matched with the start time of the next wakefulness episode.

#### 2.5.3. Splitting EEG Data According to Visual-Based Scoring Timestamps and Computing PSD Using DWT

In our study, we employed an established discrete wavelet transform (DWT) with the ‘db2’ wavelet to analyze EEG data per visual-based episode. DWT is suitable for analyzing non-stationary signals like EEG due to its optimal resolution in both time and frequency domains, allowing precise localization of time–frequency components critical for identifying specific EEG patterns related to brain activity [54,55]. It also aids artifact removal by segregating artifacts based on frequency scales, enhancing data quality, and denoising EEG signals by decomposing them into different scales and selectively reducing noise components, resulting in an improved signal–noise ratio [36,56,57]. We chose the ‘db2’ wavelet function for its established effectiveness in EEG analysis, particularly valuable in dynamic contexts [58,59]. A study on EEG signal classification showed that the Daubechies wavelets, specifically the db2 wavelet, achieved 97.2% accuracy, surpassing coeif4, sym10, db1, and db6 [60]. At level 3 decomposition, we extracted approximation and detail coefficients, representing unique frequency components within the EEG signal. These coefficients were separated into beta (15–30) Hz, alpha (7.5–15) Hz, theta (4–7.5) Hz, and delta (1–4) Hz frequency bands, as detailed in Figure 5.

Power spectral density (PSD) was computed for each band (PSD alpha, PSD theta, and PSD delta) using MATLAB’s ‘bandpower’ function to measure the signal’s power within these specific ranges, including their respective ratios (theta–alpha, delta–alpha, and delta–theta), for each episode identified through visual-based scoring. We hypothesized that during the transition from wakefulness to drowsiness, PSD alpha decreases, while PSD theta, PSD delta, and theta–alpha, delta–alpha, delta–theta ratios exhibit opposite trends. Additionally, we also calculated spectral entropy (SE), spectral spread (SS), spectral centroid (SC), and spectral rolloff (SRO) using the following computations:
SE quantifies the level of complexity or randomness present in the power spectrum of an EEG signal. A high SE value indicates a signal with high complexity and unpredictability, often associated with a wakeful state. In contrast, a low SE value suggests a more predictable and periodic signal, commonly observed during drowsiness or sleep states [61,62].
(2)$$H=-\sum_{f=0}^{L-1}n_{f}\cdot \log_{2}(n_{f})$$SE is calculated by first normalizing the spectral energy across all frequency bands. This normalization involves dividing the energy in each frequency band by the total energy across all bands. Following the normalization, SE is determined by summing the product of the normalized energy in each band and the logarithm (typically base 2) of that normalized energy. This summation is performed across all frequency bands involved in the analysis [63].SS quantifies the variability in the distribution of spectral energy within an EEG signal. It assesses the breadth of the power spectrum and reveals how energy is distributed around the spectral centroid, providing insight into the ‘sharpness’ or ‘flatness’ of the spectrum. We suggested that higher SS values are associated with drowsiness episodes, while lower values are indicative of wakefulness episodes.
(3)$$S_{i}=\sqrt{\frac{\sum_{k=1}^{WfL}{\left(k-C_{i}\right)}^{2\;}X_{i}\left(k\right)}{\sum_{k=1}^{WfL}\;X_{i}\left(k\right)}}$$SS is computed as the square root of the weighted variance of the squared differences between each frequency and the spectral centroid. It represents the standard deviation of the frequency components around the spectral centroid. This computation requires the value of $C_{i}$, the spectral centroid, to be determined first [63].SC represents the ‘center of mass’ of the power spectrum of an EEG signal. It corresponds to the average frequency of the power spectrum, weighted by the amplitude of each frequency component. We hypothesized that elevated SC values are associated with wakefulness episodes, whereas lower values tend to indicate drowsiness.
(4)$$C_{i}=\frac{\sum_{k=1}^{WfL}kX_{i}\left(k\right)}{\sum_{k=1}^{WfL}\;X_{i}\left(k\right)}$$The value of the spectral centroid, $C_{i}$, for the ith frame is computed by taking the sum of each frequency multiplied by its corresponding amplitude divided by the sum of all amplitudes where k represents the frequency index, $X_{i}\left(k\right)$ is the amplitude at frequency k, and WfL is the windowed frame length over which the computation is performed [63].SRO is the frequency below which a defined percentage (typically 85% to 95%) of the total spectral energy is contained. It is a measure used to describe the skewness of the power spectrum. We proposed that higher SRO values are linked with wakefulness, whereas lower values suggest drowsiness.
(5)$$\sum_{k=1}^{m}X_{i}\left(k\right)=C\;\sum_{k=1}^{WfL}X_{i}\left(k\right)$$SRO for the ith frame is calculated by identifying the frequency bin, m, such that the cumulative sum of amplitudes up to frequency bin m is equal to a percentage C of the total sum of amplitudes, where C is the rolloff percentage (e.g., 0.9 for 90%) [63].

The objective of computing the aforementioned ten EEG features was to pinpoint the most robust correlation between EEG patterns and visual-based scoring by examining these features. To achieve this, we established ten separate comparative criteria for each feature. These criteria aimed to identify whether a feature’s value during a drowsiness event exceeded or fell below neighboring wakefulness episodes, as detailed in Table 2. 

This analytical process, encompassing steps labeled ‘2.5.1’ to ‘2.5.3’, was executed on EEG data across all channels for each participant. This methodology was then replicated across all fifty subjects within our study cohort. We categorized visual-based drowsiness and wakefulness episodes based on adherence to these established criteria. Episodes meeting the criteria were classified as indicating a correlation, while those not conforming were deemed indicative of a lack of correlation, as depicted in Figure 6. Employing these predefined criteria, the algorithm evaluated Spearman’s correlation between episodes from visual-based scoring and instances where individual EEG features matched with these episodes across all channels. Spearman’s correlation evaluates the relationship between two variables using a monotonic function, which suits our data that do not meet the normality assumption [64]. Figure 7 shows the concurrent analysis of visual-based scoring and EEG patterns (theta–alpha ratio) captured by channel F4. Notably, episodes 5, 16, 17, 24, 25, and 26 did not meet the established criteria, thus indicating a lack of correlation. 

### 2.6. Sensitivity of EEG Channels and Brain Regions in Correlating Visual-Based Scoring with EEG Patterns

Our hypothesis aimed to assess the sensitivity across individual EEG channels and brain regions in establishing correlations between visual-based scoring and EEG patterns, characterized by ten distinct features, across fifty drivers. Our goal was to pinpoint the feature that exhibits the strongest correlation and determine which specific EEG channel or brain region contributes most significantly to this correlation. To assess this, we initially computed the average sensitivity (refer to Equation (7)) of individual EEG channels in detecting this correlation across a cohort of fifty drivers based on each feature’s comparative criterion. Next, we evaluated the average combine sensitivity of paired EEG channels (F4/F3, C4/C3, and O1/O2) and then the average combine sensitivity of all EEG channels for each feature, using the same cohort and criterion (refer to Equation (9)).
(6)$$\mathrm{Sensitivity}\;\mathrm{of}\;a\;\mathrm{Channel}=\frac{\mathrm{Episodes}\;\mathrm{showing}\;\mathrm{correlation}\;}{\mathrm{Total}\;\mathrm{Number}\;\mathrm{of}\;\mathrm{episodes}}*100$$
(7)$$\mathrm{Average}\;\mathrm{Sensitivity}=\frac{\mathrm{Sum}\;\mathrm{of}\;\mathrm{sensitivty}\;\mathrm{of}\;a\;\mathrm{channel}\;\mathrm{across}\;\mathrm{all}\;\mathrm{subjects}\;}{\mathrm{Total}\;\mathrm{number}\;\mathrm{of}\;\mathrm{subjects}\;}$$
(8)$$\mathrm{Combine}\;\mathrm{Sensitivity}\;=\left[1-\frac{\mathrm{Events}\;\mathrm{not}\;\mathrm{correlated}\;\mathrm{by}\;\mathrm{merging}\;\mathrm{channels}}{\mathrm{Total}\;\mathrm{number}\;\mathrm{of}\;\mathrm{episodes}}\right]*100$$
(9)$$\mathrm{Average}\;\mathrm{Combine}\;\mathrm{Sensitivity}\;\;=\frac{\mathrm{Sum}\;\mathrm{of}\;\mathrm{combine}\;\mathrm{sensitivity}\;\mathrm{across}\;\mathrm{all}\;\mathrm{subjects}\;}{\mathrm{Total}\;\mathrm{number}\;\mathrm{of}\;\mathrm{subjects}\;}$$

## 3. Results

### 3.1. Significant Correlation between Visual-Based Scoring and EEG Patterns across All Channels

The concurrent analysis successfully validated visual-based scoring by establishing one-to-one correlations between synchronous EEG patterns, characterized by ten specific features, and episodes of drowsiness and wakefulness derived from it. Among 927 visual-based scoring episodes, 878 matched with EEG patterns across all channels, thereby enhancing the reliability of drowsiness detection (see Table 3). Although all EEG features displayed statistically significant correlations, as demonstrated in Table 4, the theta–alpha ratio exhibited a stronger association (r = 0.9971, *p* < 0.001) with visual-based scoring compared to other analyzed features. Furthermore, we observed variations in this correlation by altering the number of channels, their positions around the head, and distinct EEG features, as shown in Figure 8.

### 3.2. Enhanced Sensitivity of F4 and O2 Channels and Frontal and Occipital Brain Regions in Correlating Visual-Based Scoring with EEG Patterns

We evaluated the average sensitivity of individual EEG channels and distinct brain regions in correlating visual-based scoring episodes with ten specific EEG features across fifty drivers. Notably, channel F4 exhibited a higher average sensitivity (75.4%) in establishing the correlation between visual-based scoring and the EEG feature (theta–alpha ratio), as demonstrated in Figure 9. Table 5 illustrates the channel with heightened sensitivity compared to all others for each EEG feature. No central channel (C3 or C4) showed higher sensitivity for any EEG feature. Similarly, the frontal brain region displayed higher average combine sensitivity (86.4%) in depicting the correlation between episodes of visual-based scoring with the theta–alpha ratio compared to other regions, as seen in Figure 10. Table 6 presents the brain region with increased sensitivity compared to all other areas for each EEG feature. An extensive analysis across all channels demonstrated that the theta–alpha ratio exhibited the highest average combine sensitivity (94.7%) in correlation with visual-based scoring, as shown in Figure 11. Furthermore, other metrics, including the spectral spread (87.8%), spectral centroid (87.4%), delta–alpha ratio (87.2%), delta–theta ratio (86.7%), and spectral rolloff (86.4%), also displayed notable sensitivity. Figure 12 illustrates the variability in sensitivity across channel F4, the frontal brain region, and all channels among fifty subjects considering theta–alpha ratio. 

## 4. Discussion

To address limitations in PERCLOS computation, we employed adaptive thresholding for eye aspect ratio calculation and to enhance the reliability of visual-based scoring, we established a one-to-one correlation between episodes derived from visual-based scoring and corresponding EEG patterns, categorized by ten distinct features.

Studies [46,47,48,49,50] computed EAR by employing a fixed threshold method prone to inaccuracies due to factors like image quality, eyewear interference, lighting variations, and head movements [26,27]. To address these limitations, we applied adaptive thresholding by fine tuning the parameters of the median filter, moving average filter, and subtracting a constant value. Before implementing it on our dataset, we validated its performance using publicly available datasets (eyeblink8 and TalkingFace) [65]. Eyeblink8 comprises eight videos, including footage of one individual wearing glasses among four participants, while TalkingFace involves a single video primarily featuring a person facing the camera with slight variations that may pose challenges for precise eye detection. Our adaptive thresholding technique demonstrated its capability by accurately detecting 365 out of 399 actual eye blinks [66]. In our current study, it successfully detected eye blinks that were overlooked by the fixed threshold [48] methodology and recorded 453 episodes of drowsiness and 474 episodes of wakefulness. To further enhance the reliability of drowsiness detection, we effectively correlated it with physiological signals, specifically EEG patterns—an advancement not explored in prior studies [18,19,29,30,31]. A total of 427 (94.3%) episodes of drowsiness matched with EEG patterns. Likewise, 451 (95.1%) episodes of wakefulness paired with EEG patterns. 

A couple of studies observed heightened theta [27] and delta [27,67] brain activities when drivers transition from wakefulness to drowsiness. Other research indicated that drowsiness is often linked to decreased EEG activity and marked by increased theta frequency band dominance [9,68] or a decline in alpha activity, especially evident when eyes are closed [69]. In our previous study [36], we noticed an increase in theta–alpha ratio during microsleep episodes and a decrease during wakefulness events, mirroring the trend found in [70]. A couple of studies [61,62] found higher mean spectral entropy values during wakefulness compared to periods of increased sleepiness. Leveraging these findings, we calculated alpha, theta, and delta power values, as well as their respective ratios and additional spectral features for episodes of drowsiness and wakefulness derived from visual-based scoring. Our study revealed a consistent pattern across various EEG features: a decrease in alpha activity during drowsiness and an increase during wakefulness episodes (708 out of 927 events), mirrored by a similar trend observed in spectral entropy (551 out of 927 events). Correspondingly, an increase in theta activity was noted during drowsiness, contrasted by a decrease during wakefulness episodes (712 out of 927 events), which echoed a parallel shift in delta activity (761 out of 927 events). Notably, the theta–alpha ratio highlighted superior performance among all features analyzed, displaying an increase during drowsiness and a decrease during wakefulness episodes (878 out of 927 events). These findings, consistent with prior studies [36,66,67], underscore a strong alignment between these EEG features and visual-based scoring across all drivers. Additionally, introducing new parameters enriched our analysis: the delta–alpha ratio mirrored a similar trend to delta–theta, increasing during drowsiness and decreasing during wakefulness episodes (808 and 804 out of 927 events, respectively). Furthermore, the spectral centroid and rolloff both demonstrated a decrease during drowsiness and an increase during wakefulness episodes (811 and 802 out of 927 events, respectively). In addition, spectral spread indicated an increase during drowsiness and remained elevated during wakefulness episodes (814 out of 927 events).

Our study also aimed to determine the optimal count of individual channels and brain regions sensitive to correlations between visual-based scoring and distinct EEG features. A couple of studies [36,71] have identified that the frontal brain region exhibits higher sensitivity in detecting changes within the theta and alpha frequency bands as a driver transitions from an awake to a drowsy state. In our study, channel F4 and the frontal region exhibited superior sensitivity in detecting variations in the theta–alpha ratio and theta activity during the transition from wakefulness to drowsiness, surpassing other individual channels and brain regions. Our analysis also highlighted that channel O2, along with the occipital brain region, consistently demonstrated heightened sensitivity across various EEG features, particularly in detecting alpha and delta activity. These findings align with previous research that highlighted the significant correlation between EEG alterations in the occipital region and levels of driver drowsiness [72,73]. Furthermore, another study [74] established a direct association between eye closure degree (ECD) and occipital alpha activity. Our analysis also revealed that the central brain region did not demonstrate superiority across the analyzed features.

In contrast to prior studies [28,75,76] that defined drowsiness at PERCLOS thresholds of ≥0.15 and ≥0.20, our research employs a higher PERCLOS threshold of ≥0.30 to better accommodate unusual blinking patterns. To enhance the reliability of our drowsiness detection, we integrated visual-based assessments with EEG patterns, ensuring that instances flagged as potential drowsiness due to abnormal blinking are correctly classified as wakefulness instead of drowsiness based on EEG patterns. For example, while visual-based scoring identified 453 episodes as drowsiness, upon validation with EEG patterns, 427 episodes were found to align with EEG patterns.

Studies have reported high accuracy in correlating PERCLOS measurements with various physiological and driving-related parameters: a study achieved 92.5% accuracy by linking heart rate variability-derived parameters, a couple of studies [32,33] found an average accuracy of 90.7% and 94% by associating PERCLOS with driving lane positions and steering wheel movement, and yet another [34] determined 91% accuracy by linking PERCLOS with galvanic skin response (GSR). In contrast, our study demonstrated a higher average sensitivity (94.7%) between episodes of visual-based scoring and EEG patterns (theta–alpha-ratio).

Although recent machine learning and deep learning methodologies have contributed to accurately classifying EEG signals [77,78,79], we intentionally avoided them in our research for several reasons. Firstly, our study segmented EEG data based on visual-based episodes. Given the intrinsic variability in the duration of these episodes, both within and across subjects, the length of EEG segments was not uniform. Consequently, employing a fixed-length window for EEG signal segmentation, as is common in deep learning frameworks, was not feasible [53,80]. To accommodate the heterogeneous EEG segment lengths and preserve the integrity of subject-specific EEG patterns, we devised a novel approach. Secondly, we aimed to establish one-to-one correlations using selected EEG features instead of employing a black-box approach inherent in deep learning methods. Lastly, due to the limited data set consisting of 927 episodes (474 wakefulness and 453 drowsiness), using deep learning methods during training could potentially lead to overfitting [81,82]. Our methodology has demonstrated additional benefits that cannot be achieved using deep learning methods, as depicted in Table 7.

### Limitations of the Study and Future Perspective

Our deliberate choice to exclusively recruit male drivers was methodically justified in the ‘Study Population’ section. While humidity levels were not directly monitored, we believe their impact on driving drowsiness to be minimal given the controlled temperature environment. As both PERCLOS and EEG can effectively detect the onset of drowsiness, we refrained from integrating them with driving attributes due to the potential delay between driving cues and the onset of drowsiness [27]. Importantly, certain studies [85,86] have highlighted instances of drowsiness or microsleeps occurring with open eyes, rendering these episodes unidentifiable through visual-based scoring methods. We also avoided the manual interpretation of EEG patterns due to their highly challenging nature, which is prone to human error and labor-intensive [9,36]. The utilization of only two frontal and two occipital EEG channels in our study may pose a potential limitation in not matching all episodes of visual-based scoring with EEG patterns, as our findings suggest that augmenting the number of frontal or occipital channels within their respective regions notably enhances this correlation. Although our study focused solely on traditional frequency bands, their ratios, and spectral features, exploring non-linear features might yield stronger correlations. Additionally, amalgamating the features into a metric might enhance the correlation. Our approach to assessing fitness to drive in OSA drivers is based on quantifying the frequency of drowsiness episodes during simulated driving rather than directly evaluating driving performance or attributes. This method indirectly contributes to understanding the fitness to drive by highlighting the potential risk posed by drowsiness episodes. In our study, we could not perform uniform normalization process across subjects, as it could obscure the unique contributions of specific channels and features within individual EEG profiles. Future research could center on enhancing frontal or occipital channels specifically, amalgamating existing features and identifying novel nonlinear EEG features to achieve greater alignment with visual-based scoring events. Moreover, integrating PERCLOS with electrooculography (EOG) may enhance visual-based scoring accuracy, with additional improvement possible by incorporating mouth and head motion-based features. Lastly, future research may develop a methodology to integrate EEG data, visual-based scoring, and vehicle-based parameters, considering the lag between EEG signals (visual-based scoring) and vehicle-based parameters.

## 5. Conclusions

Our concurrent analysis, integrating visual-based scoring episodes with EEG patterns across ten distinct features, significantly enhances the reliability of drowsiness detection through a one-to-one correlation. Additionally, our adaptive thresholding technique in PERCLOS computation mitigates the associated limitations. We determined the average sensitivity of EEG channels and brain regions across fifty drivers in correlating visual-based scoring with EEG patterns, highlighting enhanced sensitivity in specific EEG channels (F4 and O2) and brain regions (frontal and occipital). Augmenting the number of frontal or occipital channels beyond those used in this study may align all instances of visual-based scoring with their corresponding EEG patterns. Notably, among the analyzed features, the theta–alpha ratio exhibited the highest alignment with visual-based scoring, followed by the delta–alpha and delta–theta ratios, respectively. Combining these features into a collective metric might further improve this correlation. 

Our study offers a crucial tool for healthcare professionals and road safety experts by facilitating fitness-to-drive assessments for drivers with OSA. Additionally, it establishes a framework to enhance the reliability of real-time drowsiness detection while minimizing the hardware requirements.

## Figures and Tables

**Figure 1 sensors-24-02625-f001:**
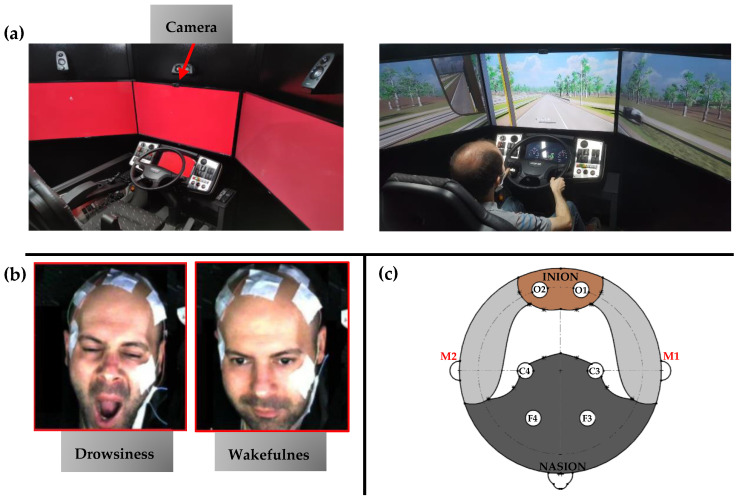
(**a**) A high-fidelity driver training simulator comprising a driver cabin, camera system, voice communication setup, acceleration and brake pedals, steering controls, offering both automatic and manual transmission modes, and providing diverse training scenarios [36]; (**b**) facial video recording conducted by a 1080p camera mounted atop the middle view screen; (**c**) the international 10–20 system utilized for EEG electrode placement on the subject’s scalp, positioning electrodes at F3 and F4, C3 and C4, O1 and O2, with M1 and M2 serving as references.

**Figure 2 sensors-24-02625-f002:**
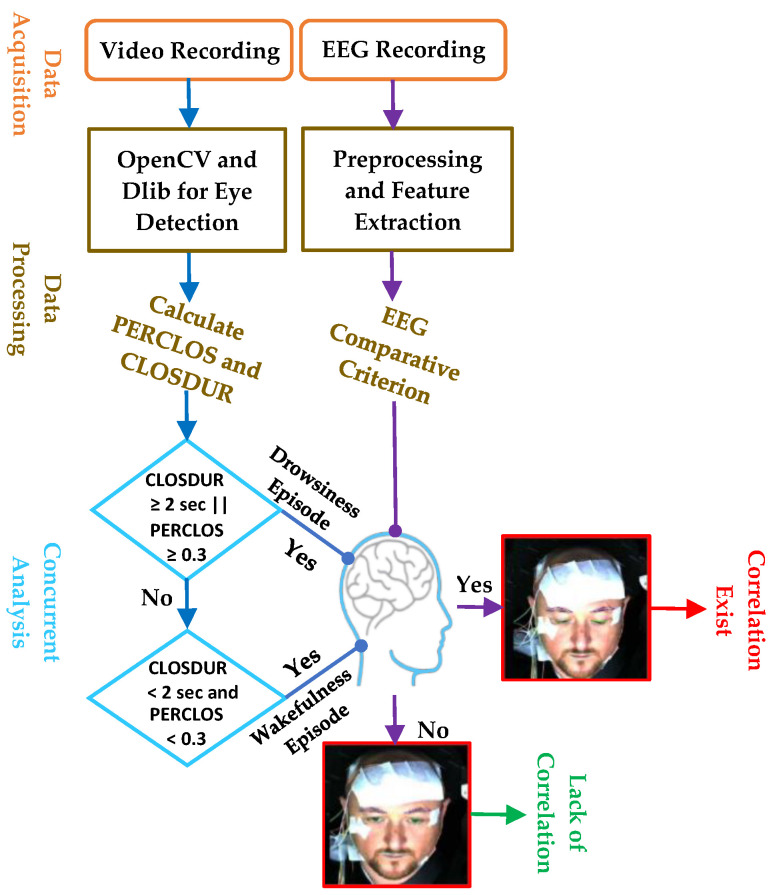
The experiment design began with the acquisition of facial videos and EEG signals, followed by data processing and feature extraction. Subsequently, a concurrent analysis was conducted to validate visual-based scoring against EEG patterns, confirming the onset of drowsiness.

**Figure 3 sensors-24-02625-f003:**
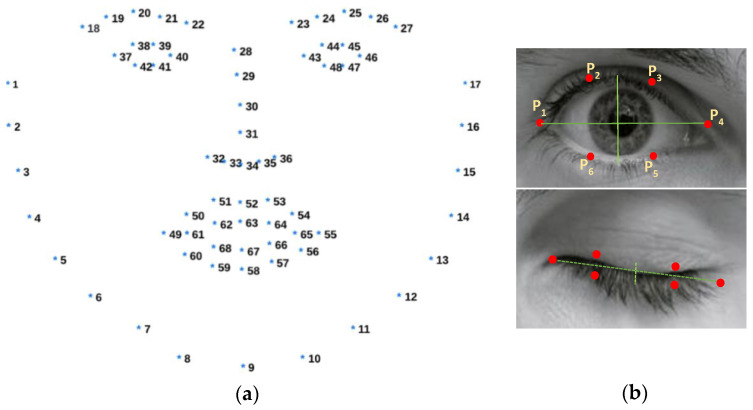
(**a**) A total of 68 facial landmark points provided by Dlib library. (**b**) Open and closed eyes with detected landmark points. These points around the eye are used to calculate EAR.

**Figure 4 sensors-24-02625-f004:**
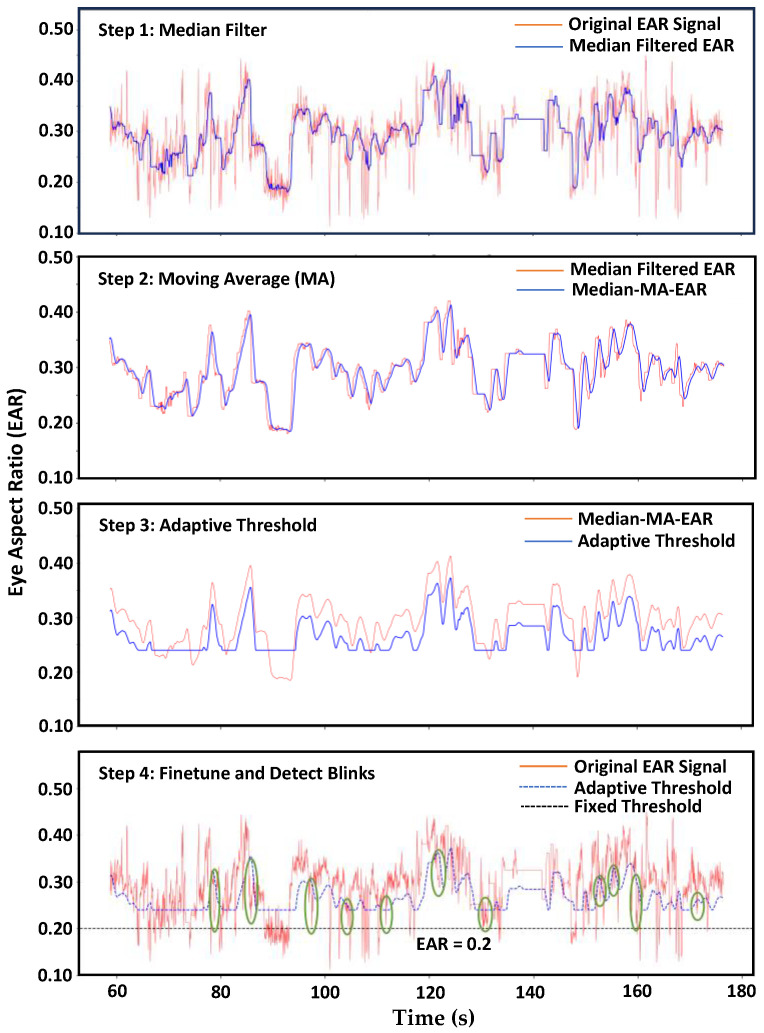
Steps of blink detection using eye aspect ratio: Following the extraction of EAR values from video frames, (**Step 1**) applies a median filter to reduce sudden and fast variations, noticeable when comparing the original signal and its median-filtered version. (**Step 2**) smoothens the signal and reduces short-term swings with a moving average filter, as demonstrated by the Median-MA-EAR signal. (**Step 3**) employs an adaptive threshold to enhance accuracy and make the signal condition adaptive. (**Step 4**) finetunes the parameters and selects consecutive signals falling below the threshold to identify blinks (green ellipses).

**Figure 5 sensors-24-02625-f005:**
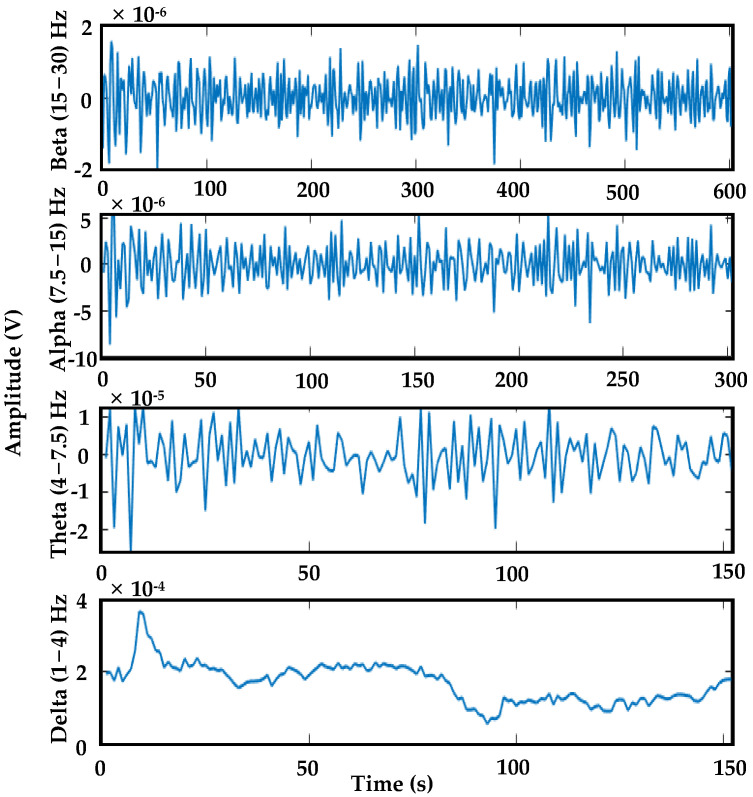
Detail and approximation coefficients associated with their respective frequency bands (beta, alpha, theta, and delta) obtained through the implementation of DWT.

**Figure 6 sensors-24-02625-f006:**
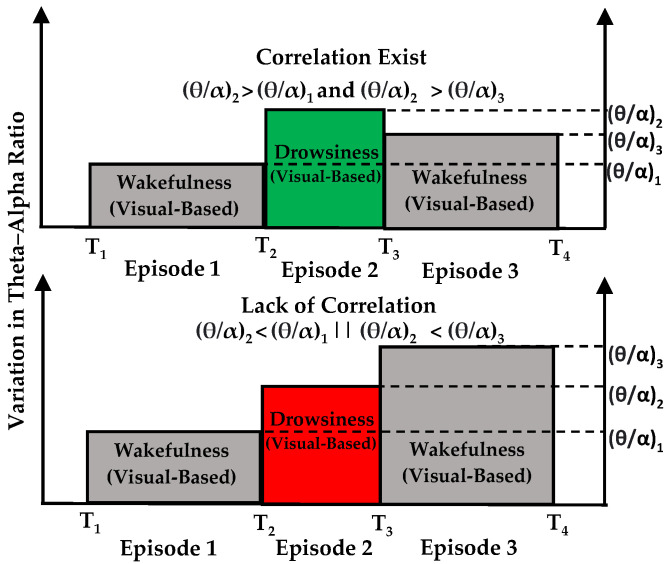
A comparative approach used to determine the presence of correlation by combining episodes from visual-based scoring with EEG patterns.

**Figure 7 sensors-24-02625-f007:**
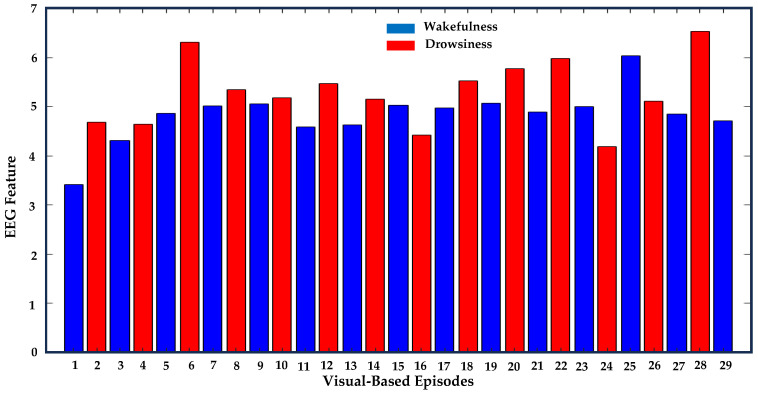
This figure presents a concurrent analysis of a participant (ID:1055). Blue and red bars represent neighboring wakefulness and drowsiness episodes determined by visual-based scoring throughout the entire driving period, with the length of the bar indicating the corresponding EEG patterns (theta–alpha ratio). In this instance, visual-based scoring recorded 15 wakefulness and 14 drowsiness events (total: 29 episodes). Comparative criterion for theta–alpha ratio reveals that EEG patterns correlate with 23 episodes of visual-based scoring, demonstrating F4-channel sensitivity of 79.3% (23/29) × 100).

**Figure 8 sensors-24-02625-f008:**
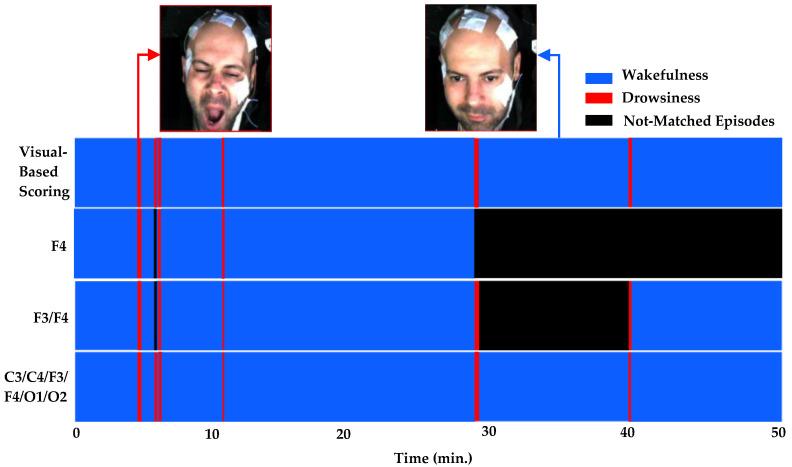
This figure depicts the matching between visual-based scoring and EEG patterns (subject ID:1025), showcasing variations with the number of channels. The top row presents visual-based scoring, encompassing six drowsiness and seven wakefulness events. Subsequent rows demonstrate the matching of these episodes with EEG patterns based on different channels: the second row with channel F4, the third by combining channels F3 and F4, and the last using all channels. Notably, all visual-based episodes corresponded with EEG patterns in the combined channel setup.

**Figure 9 sensors-24-02625-f009:**
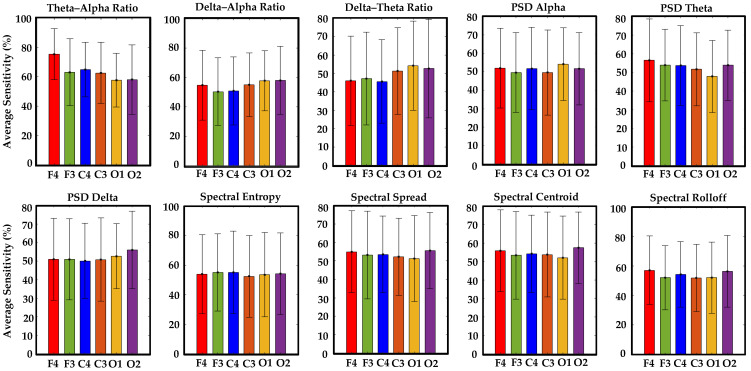
This figure shows the average sensitivity of individual EEG channels in detecting correlations between episodes of visual-based scoring and ten specific EEG features across fifty drivers. The theta–alpha ratio emerged as a crucial feature for effectively correlating EEG patterns with visual-based scoring and channels F4 and O2 maintained consistent superiority across most EEG features.

**Figure 10 sensors-24-02625-f010:**
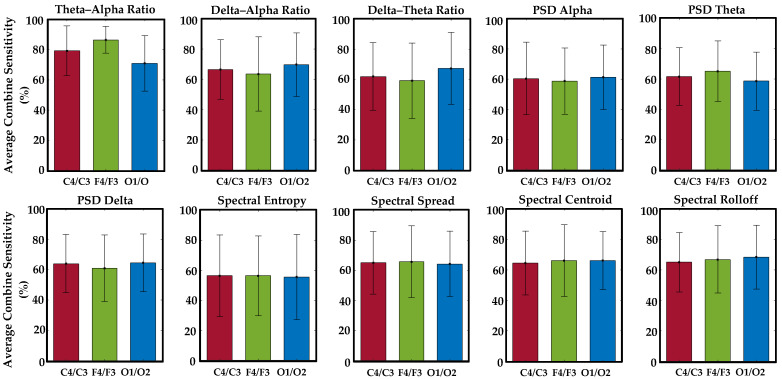
This figure illustrates the average combine sensitivity of paired EEG channels across brain regions in detecting correlations between episodes of visual-based scoring and ten specific EEG features in a cohort of fifty drivers. Notably, the frontal and occipital regions sustained consistent supremacy across most EEG features in establishing this correlation. The central region did not exhibit supremacy for any of the features.

**Figure 11 sensors-24-02625-f011:**
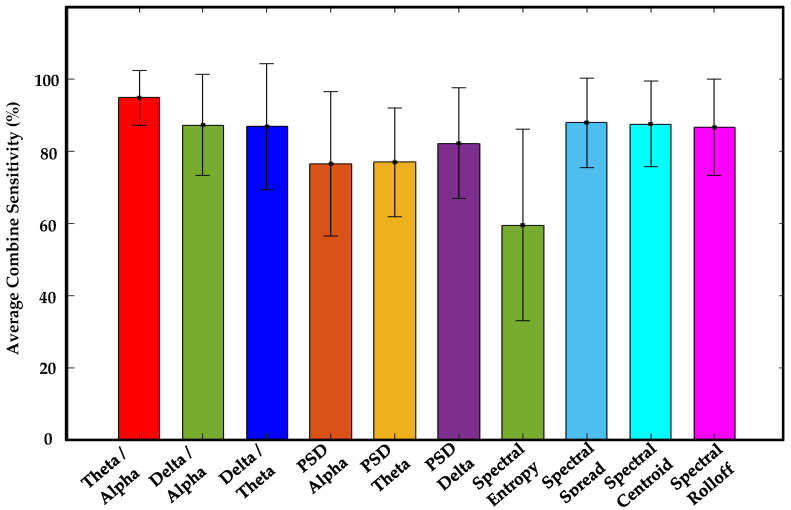
This figure illustrates the average combine sensitivity of all EEG channels (F3/F4/C3/C4/O1/O2) in detecting correlations between episodes of visual-based scoring and ten specific EEG features across a cohort of fifty drivers. Notably, all of the features except spectral entropy demonstrated average combine sensitivity of more than 75%.

**Figure 12 sensors-24-02625-f012:**
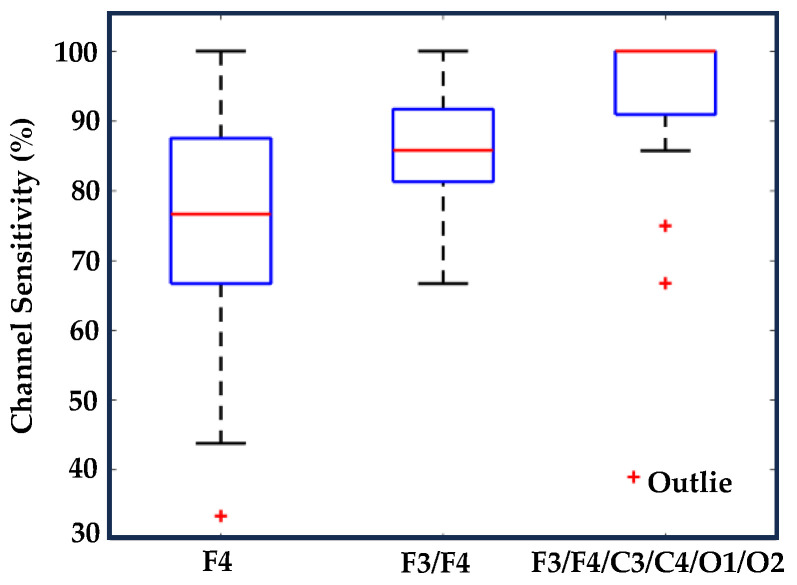
This figure illustrates the decrease in the variability of channel sensitivity with an increasing number of channels.

**Table 1 sensors-24-02625-t001:** Demographics of fifty subjects. Each row displays the minimum–maximum (mean ± standard deviation) of a corresponding characteristic.

Parameter	Value
Sex	All are males
Age	32–68 (47.9 ± 7.6) year
Body Mass Index (BMI)	23.5–41.9 (31.3 ± 4.4) kg/m^2^
Last night sleep hours	1–11 (6.3 ± 1.8) hour
Apnea–Hypopnea Index (AHI)	5–103.5 (29.8 ± 23.2)/hour
Oxygen Desaturation Index (ODI)	1.0–87.8 (24.4 ± 22.7)/hour

**Table 2 sensors-24-02625-t002:** Ten criteria illustrate distinct EEG features for correlation with visual-based scoring, where ‘i’ represents drowsiness episodes, and ‘i − 1′ and ‘i + 1′ denote the preceding and subsequent wakefulness episodes derived from visual-based scoring.

EEG Feature	Criterion for Correlation
Theta–alpha ratio	theta_alpha_ratio(i) > theta_alpha_ratio(i − 1) && theta_alpha_ratio(i + 1)
Delta–alpha ratio	delta_alpha_ratio(i) > delta_alpha_ratio(i − 1) && delta_alpha_ratio(i + 1)
Delta–theta ratio	delta_theta_ratio(i) > delta_theta_ratio(i − 1) && delta_theta_ratio(i + 1)
PSD Alpha	PSD_alpha(i) < PSD_alpha(i − 1) && PSD_alpha(i + 1)
PSD Theta	PSD_theta(i) > PSD_theta(i − 1) && PSD_theta(i + 1)
PSD Delta	PSD_delta(i) > PSD_delta(i − 1) && PSD_delta(i + 1)
Spectral Entropy	PSD_entropy(i) < PSD_entropy(i − 1) && PSD_entropy(i + 1)
Spectral Spread	PSD_spread(i) > PSD_spread(i − 1) && PSD_spread(i + 1)
Spectral Centroid	PSD_centroid(i) < PSD_centroid(i − 1) && PSD_centroid(i + 1)
Spectral Rolloff	PSD_rolloff(i) < PSD_rolloff(i − 1) && PSD_rolloff(i + 1)

**Table 3 sensors-24-02625-t003:** This table presents the number of episodes where visual-based scoring aligns with EEG patterns (theta–alpha ratio) across all channels analyzed concurrently.

Episodes	Visual-Based Scoring	Matched Episodes
Drowsiness	453	427 (94.3%)
Wakefulness	474	451 (95.1%)
Total Episodes	927	878 (94.7%)

**Table 4 sensors-24-02625-t004:** Spearman’s correlations calculated between episodes derived from visual-based scoring and instances where individual EEG features matched with these episodes across all six channels. This analysis encompassed a cohort of fifty subjects.

EEG Feature	Spearman’s Correlation
Theta–alpha ratio	r = 0.9942, *p* < 0.001
Delta–alpha-ratio	r = 0.9768, *p* < 0.001
Delta–theta-ratio	r = 0.9826, *p* < 0.001
PSD Alpha	r = 0.9757, *p* < 0.001
PSD Theta	r = 0.9633, *p* < 0.001
PSD Delta	r = 0.9777, *p* < 0.001
Spectral Entropy	r = 0.9268, *p* < 0.001
Spectral Spread	r = 0.9816, *p* < 0.001
Spectral Centroid	r = 0.9843, *p* < 0.001
Spectral Rolloff	r = 0.9826, *p* < 0.001

**Table 5 sensors-24-02625-t005:** This table illustrates the heightened average sensitivity of a single EEG channel for each EEG feature. Additionally, it presents the trend depicting how each EEG feature varies with increasing drowsiness.

EEG Feature	EEG Channel	Average Sensitivity	Trend
Theta–alpha ratio	F4	75.4%	↑
Delta–alpha-ratio	O2	58.0%	↑
Delta–theta-ratio	O1	54.2%	↑
PSD Alpha	O1	54.2%	↓
PSD Theta	F4	56.5%	↑
PSD Delta	O2	56.1%	↑
Spectral Entropy	F3	55.1%	↓
Spectral Spread	O2	55.6%	↑
Spectral Centroid	O2	57.5%	↓
Spectral Rolloff	F4	57.0%	↓

**Table 6 sensors-24-02625-t006:** This table illustrates the heightened average combine sensitivity of a brain region for each EEG feature. Notably, the theta–alpha ratio significantly matched with visual-based scoring in the frontal brain region across all fifty subjects.

EEG Feature	Brain Region	Average Combine Sensitivity
Theta–alpha ratio	Frontal	86.4%
Delta–alpha-ratio	Occipital	69.7%
Delta–theta-ratio	Occipital	67.3%
PSD Alpha	Occipital	61.3%
PSD Theta	Frontal	65.1%
PSD Delta	Occipital	64.1%
Spectral Entropy	Frontal	56.3%
Spectral Spread	Frontal	65.6%
Spectral Centroid	Occipital	66.1%
Spectral Rolloff	Occipital	68.4%

**Table 7 sensors-24-02625-t007:** Analytical comparison of contemporary drowsiness detection approaches utilizing deep learning and our methodology.

Study Reference	Sensing Method	Methodology	Findings and Limitations
Safarov F et al. [83]	Camera	Threshold + DL-Based	Accuracy: 95.8% Not validated with physiological signal
Bajaj, J.S. et al. [34]	Camera + Galvanic Skin Response (GSR)	MTCNN	Accuracy: 91% (GSR) is less reliable than EEG for detecting drowsiness
Arefnezhad, S. et al. [27]	SmartEye + EEG Electrodes	Encoder–Decoder Architecture	Generalized correlation between EEG patterns and PERCLOS up to 70% Cost ineffective
Arefnezhad, S. et al. [23]	Vehicle-Based	CNN + RNN	Accuracy: 96% Not validated with physiological signal
Wang, F et al. [84]	EEG Electrodes	CNN	Accuracy: 91.5% Only EEG signals were used to detect drowsiness
Our Study	Camera + EEG Electrodes	One-to-one correlation	Validation of PERCLOS with EEG patterns Correlation up to 94.7% Explored the sensitivity of different EEG channels Subject-specific approach

## Data Availability

Data are unavailable due to ethical restrictions.

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
