# Peer review of "Association of Visual-Based Signals with Electroencephalography Patterns in Enhancing the Drowsiness Detection in Drivers with Obstructive Sleep Apnea"

_sensors, 2024, doi:10.3390/s24082625_

Round 1

Reviewer 1 Report

Comments and Suggestions for Authors

1. The application details of DWT in Section 2.5.3 should be supplemented, such as the choice of  wavelet function.

2. The technical details of FIR filters in Section 2.5.1 are ignored.

3. The reasons for using Pearson correlation should be explained.

4. EEG features in Table 2 should be defined in detail, including their computation way.

5. EEG patters should be shown and introduced.

6. During experiments, deep learning methods should be compared to support the conclusion of the paper.

7. Some important related work like deep learning should be cited and analyzed since they have achieved great progress in signal or image processing, including Fine-grained modulation classification using multi-scale radio transformer with dual-channel representation, IEEE Communications Letters. Recognition of Drivers’ Hard and Soft Braking Intentions Based on Hybrid Brain-Computer Interfaces.  Study on a Portable Electrode Used to Detect the Fatigue of Tower Crane Drivers in Real Construction Environment. IEEE Transactions on Instrumentation and Measurement.

Comments on the Quality of English Language

Minor editing of English language required.

Reviewer 2 Report

Comments and Suggestions for Authors

Comments and Suggestions for Authors

This is a very interesting study. There are problems with the manuscript that made it impossible to meet publication requirements. Some major issues that the paper should be addressed as follows:

1The experiment involved the study of the closing characteristics of the eyes of the subjects, and whether the author had considered the interference of the subjects such as abnormal blinking.

2In Section 3.2 of the manuscript, the expression of the figure and table is relatively simple, please elaborate on this part of the table.

3There are many factors affecting driving fatigue, and the author needs to give a detailed description of the experimental environment, such as laboratory temperature, humidity, and lighting conditions.

4The diagrams in the manuscript should be in an editable format; The chart cannot be distorted when enlarged.

5The supporting literature on EEG research in the manuscript is not in recent years, such as reference [16], it is suggested to add the following recent studies.

[1] El-Nabi S A, El-Shafai W, El-Rabaie E S M, et al. Machine learning and deep learning techniques for driver fatigue and drowsiness detection: a review[J]. Multimedia Tools and Applications, 2024, 83(3): 9441-9477.

[2] Wang F, Gu T, Yao W. Research on the application of the Sleep EEG Net model based on domain adaptation transfer in the detection of driving fatigue[J]. Biomedical Signal Processing and Control, 2024, 90: 105832.0.

6Although the author mentioned in the discussion section that only male subjects were selected for this study, it is not enough. Do the authors need to elaborate on why only male subjects were selected and what factors were involved? Otherwise, the study is flawed.

Comments on the Quality of English Language

The manuscript makes it easy for the reader to understand the meaning of the expression.

Reviewer 3 Report

Comments and Suggestions for Authors

This study demonstrates that a method based on the detection of eye closure (percentage and duration of eye closure PERCLOS+CLOSDUR) allows the detection of episodes of drowsiness with good agreement with EEG, especially the theta-alpha ratio, in simulated driving situations.

This study is interesting in demonstrating the efficiency in the detection of drowsiness episodes based on visual-based signals such as eye closure. It also confirms that the theta-alpha ratio is a good marker of drowsiness.

Some aspects, however, need clarification.

Major :

1. In the abstract and the conclusion, the authors state that their work “enhances real-time drowsiness detection reliability and assesses fitness-to-drive in OSA drivers

If the results show the good correlation between visual-based signals and EEG patterns, no results on the association between their visual-based method (or EEG signal)

and driving performances are shown. The fitness-to-drive is then not assessed in this paper.

The participants being recorded while driving in a simulator, those analyses could have been performed and added to the manuscript, enabling to substantially increase the interest of the paper.

Even if some delay between visual-based signals and driving performances may occur, as the authors suggested l.631, association between visual-based drowsiness episodes and driving performances could be studied, within the same episodes or analyzing the driving performances in the few seconds after each drowsiness episode.

Minor:

2. Methods for EEG analyses:

Are there any artifact rejection performed, for example when movements occur?

Is there any normalization of the PSD enabling comparison between subjects?

3. What are the descriptive statistics of the drowsiness and wakefulness episodes, especially in term of duration (mean per episode, SD, total duration)?

4. from l.585 to l.593 in the Discussion section: doesn't this part belong more to the Results? The statistic description of the EEG pattern (PSD, …) between drowsiness and wakefulness episodes could be interesting to show.

5. As a complement on the theta-alpha ratio: the interest of the theta-alpha ratio to detect drowsiness has also been shown outside the context of driving, cf. this very recent review paper on sleep onset: https://www.sciencedirect.com/science/article/pii/S0166223624000183

Round 2

Reviewer 1 Report

Comments and Suggestions for Authors

All the comments are well revised. Accept.

Reviewer 2 Report

Comments and Suggestions for Authors

All the issues raised have been resolved.

Reviewer 3 Report

Comments and Suggestions for Authors

The authors have satisfactorily addressed my questions